# Calcitonin Related Polypeptide Alpha Mediates Oral Cancer Pain

**DOI:** 10.3390/cells12131675

**Published:** 2023-06-21

**Authors:** Nguyen Huu Tu, Kenji Inoue, Parker K. Lewis, Ammar Khan, Jun Hyeong Hwang, Varun Chokshi, Branka Brukner Dabovic, Shanmugapriya Selvaraj, Aditi Bhattacharya, Zinaida Dubeykovskaya, Nathalie M. Pinkerton, Nigel W. Bunnett, Cynthia A. Loomis, Donna G. Albertson, Brian L. Schmidt

**Affiliations:** 1Department of Oral and Maxillofacial Surgery, Translational Research Center, New York University College of Dentistry, New York, NY 10010, USA; n.h.tu@nyu.edu (N.H.T.); ki630@nyu.edu (K.I.); ak8108@nyu.edu (A.K.); jh7102@nyu.edu (J.H.H.); vvc2004@nyu.edu (V.C.); ab7186@nyu.edu (A.B.); zd10@nyu.edu (Z.D.); 2Department of Chemical and Biomolecular Engineering, New York University Tandon School of Engineering, New York, NY 10010, USA; pkl9458@nyu.edu (P.K.L.); nmp2@nyu.edu (N.M.P.); 3Department of Pathology, NYU Langone Health, New York, NY 10010, USA; dabov01@nyu.edu (B.B.D.); priya.selvaraj@nyulangone.org (S.S.); cindy.loomis@nyulangone.org (C.A.L.); 4Department of Molecular Pathobiology, New York University College of Dentistry, New York, NY 10010, USA; nwb2@nyu.edu; 5Department of Neuroscience and Physiology, Neuroscience Institute, NYU Langone Health, New York, NY 10016, USA; 6NYU Pain Research Center, New York University College of Dentistry, New York, NY 10010, USA

**Keywords:** oral cancer, pain, CGRP, peptidergic neurons, CALCRL/RAMP1, cancer innervation

## Abstract

Oral cancer patients suffer pain at the site of the cancer. Calcitonin gene related polypeptide (CGRP), a neuropeptide expressed by a subset of primary afferent neurons, promotes oral cancer growth. CGRP also mediates trigeminal pain (migraine) and neurogenic inflammation. The contribution of CGRP to oral cancer pain is investigated in the present study. The findings demonstrate that CGRP-immunoreactive (-ir) neurons and neurites innervate orthotopic oral cancer xenograft tumors in mice. Cancer increases anterograde transport of CGRP in axons innervating the tumor, supporting neurogenic secretion as the source of CGRP in the oral cancer microenvironment. CGRP antagonism reverses oral cancer nociception in preclinical oral cancer pain models. Single-cell RNA-sequencing is used to identify cell types in the cancer microenvironment expressing the CGRP receptor components, receptor activity modifying protein 1 *Ramp1* and calcitonin receptor like receptor (CLR, encoded by *Calcrl*). *Ramp1* and *Calcrl* transcripts are detected in cells expressing marker genes for Schwann cells, endothelial cells, fibroblasts and immune cells. *Ramp1* and *Calcrl* transcripts are more frequently detected in cells expressing fibroblast and immune cell markers. This work identifies CGRP as mediator of oral cancer pain and suggests the antagonism of CGRP to alleviate oral cancer pain.

## 1. Introduction

Oral cancer is a subset of head and neck cancer. Oral cancer is distinct amongst cancers in that it is associated with the highest prevalence and greatest severity of pain of all cancers [1]. Oral cancer pain is generated in the periphery within the cancer microenvironment [2]. Evidence for a peripheral mechanism is supported by the finding that surgical removal of the cancer reverses the pain, and clinical experience that local anesthetic injection is temporarily effective for reversing the pain [3]. Despite these clinical findings our understanding of the role of afferent and efferent innervation and the associated neurotransmitters and receptors is poorly understood.

The effect of sympathetic innervation on cancer proliferation is known for head and neck cancer. Bilateral cervical sympathectomy resulted in smaller and less invasive oral cancers that were induced by 4-nitroquinoline 1-oxide (4NQO) [4]. A similar proliferative effect of sympathetic innervation on prostate cancer was observed following surgical transection of the hypogastric nerve [5], and chemical disruption of sympathetic innervation reduces breast cancer growth [6]. The report of pain by cancer patients requires the activity of afferent neurons. Head and neck cancers induce sprouting of TRPV1-positive sensory nerves, but not autonomic neurons [7]. Oral cancers, however, can induce the reprogramming of afferent sensory neurons to exhibit sympathetic activity [8]. The conversion of afferent neurons to a sympathetic phenotype requires the loss of the tumor suppressor gene *TP53* and the loss of the microRNA miR-34a. Pathologic evaluation of oral cancers resected from patients showed increased neuronal density of nerves expressing tyrosine hydroxylase, a molecular marker of sympathetic neurons. Sympathetic innervation to the oral cavity is efferent activity, while afferent activity is responsible for the sensation of pain. 

Neuronal activity by a subset of afferent neurons known as peptidergic neurons is responsible for neurogenic inflammation. Activation of peptidergic neurons results in the secretion of neuropeptides, substance P and calcitonin gene-related peptide (CGRP). The effects of CGRP result in tissue changes including edema and erythema, as well as pain. CGRP is a 37 amino acid peptide produced via alternative splicing of the *CALCA* transcript. The CGRP receptor comprises two membrane components: receptor activity-modifying protein 1 (RAMP1), a single transmembrane domain chaperone, and calcitonin receptor-like receptor (CLR, encoded by *CALCRL*), a G-protein coupled-receptor (GPCR). A third component, the CGRP receptor component (CRCP), provides coupling to G_αs_ to activate adenylate cyclase and production of cAMP. 

Oral and oropharyngeal cancers are innervated by CGRP-ir neurons, which modulate tumor infiltrating immune cells [9] or promote cancer cell survival in nutrient-poor conditions [10]. CGRP expression by oral cancers correlates with worse overall survival [10,11] and biologic activity. For example, CGRP plasma levels correlated with perineural invasion, a process whereby the cancer infiltrates the perineural space, and lymph node metastasis [11]. Metastasis significantly correlates with self-reported oral cancer pain [12]. CGRP is well-established to induce pain in migraine—a form of headache mediated by peripheral trigeminal sensory neurons [13,14]. 

The contribution of CGRP to oral cancer pain is unclear. A small study of oral and oropharyngeal cancer patients found a trend for the association of pain with the presence of CGRP-ir nerves in the cancer microenvironment [9]. The study, however, included patients with oral cancer and oropharyngeal cancer, which are biologically and clinically distinct cancers. Here, innervation of oral cancers by primary afferent neurons containing CGRP was investigated, and whether antagonizing CGRP could attenuate pain in preclinical models was examined. Cell types in the oral cancer microenvironment likely to be responsive to CGRP because they express CGRP receptors were identified.

## 2. Materials and Methods

### 2.1. Cell Lines

Human oral tongue cancer cell lines, HSC-3 (JCRB Cat# JCRB0623, RRID:CVCL_1288, passages 4–12) and OSC-20 (Cat# JCRB0197, RRID:CVCL_3087, passages 4–12) were obtained from the Japanese Collection of Research Bioresources Cell Bank, Osaka, Japan. The spontaneously immortalized human keratinocyte cell line HaCaT (RRID:CVCL_0038, passages 4–12) was purchased from AddexBio Technologies (Cat# T0020001, San Diego, CA, USA). Human dysplastic keratinocyte cell line DOK (RRID:CVCL_1180, passages 4–12) was purchased from Sigma-Aldrich (Cat# 94122104, St. Louis, MO, USA). The human cell lines were cultured in Dulbecco’s modified Eagle’s medium (DMEM, Gibco, Waltham, MA, USA) supplemented with 10% fetal bovine serum (FBS) and 1% penicillin/streptomycin (50 U/mL). Mouse cell line MOC2 (RRID:CVCL_ZD33, passage 5), obtained from Dr. R. Uppaluri [15], was derived from a primary tumor induced by dimethylbenz(a)anthracene (DMBA) carcinogen treatment in a Cxcr3 deficient mouse on a C57BL/6 background. The MOC2 cells are aggressive and metastasize to regional lymph nodes. The MOC2 cells were cultured according to the recommended protocol provided by Kerafast, Inc. (Boston, MA, USA). 

### 2.2. Multiplex Immunofluorescence Imaging of Cancer Innervation

#### 2.2.1. Generation of Mouse Orthotopic Xenografts

Cancer cell lines HSC-3 and OSC-20 were cultured to 70–80% confluence. Collected HSC-3 (1 × 10^5^ cells) or OSC-20 (2 × 10^5^ cells) cells with viability 95–98% were mixed in a 1:1 ratio with Matrigel (Cat# 354234, Corning, Somerville, MA, USA,) and 20 µL of the mixture was injected into the left lateral tongue of nude mice (NU/J, Jackson Laboratories strain #:002019, RRID:IMSR_JAX:002019, Bar Harbor, ME, USA), which are required for generation of a xenograft, under anesthesia. Mice were sacrificed on post-inoculation day 30. The tongues were harvested, bisected longitudinally and fixed in 10% neutral buffered formalin. Tissues were dehydrated through graded ethanols and xylene and then processed into paraffin (Paraplast X-tra, Cat# 39603002, Leica Biosystems, Deer Park, IL, USA) on a Leica Peloris II tissue processor. Embedded tissues were sectioned at 5 µm and every 10th slide with two sections per slide stained with hematoxylin (Cat# 3801575, Leica Biosystems, Deer Park, IL, USA) and eosin (Cat# 3801619, Leica Biosystems, Deer Park, IL, USA) on a Leica ST5020 automated stainer to evaluate histology. Animal experiments were carried out in accordance with the recommendations of the National Institute of Health guidelines and the PHS Policy on the Humane Care and Use of Laboratory Animals. The protocol (IA16-00437) was approved by the New York University Langone Health Institutional Animal Care and Use Committee.

#### 2.2.2. Multiplex Immunofluorescence Staining and Imaging of Mouse Orthotopic Xenografts

Adjacent tumor-containing sections were stained with Akoya Biosciences^®^ Opal™ (Marlborough, MA, USA) multiplex immunofluorescence reagents on a Leica BondRx autostainer, according to the manufacturers’ instructions. In brief, the slides were incubated with the first pair of primary antibody (Appendix A) and Rabbit-on-Rodent HRP-Polymer secondary (SKU: RMR622L, Biocare Medical, Pacheco, CA, USA) followed by HRP-mediated tyramide signal amplification with a specific Opal^®^ fluorophore. The primary and secondary antibodies were subsequently removed with a heat retrieval step, leaving the Opal fluorophore covalently linked to the antigen. This sequence was repeated with a subsequent primary/secondary antibody pair and a different Opal fluorophore at each step. The sections were counterstained with spectral DAPI (SKU: FP1490Akoya Biosciences, Marlborough, MA, USA) and mounted with ProLong Gold Antifade (Cat# P36935, ThermoFisher Scientific, Waltham, MA, USA). Semi-automated image acquisition was performed on an Akoya Vectra Polaris/PhenoImager. After whole slide scanning at 20× in the Motif mode, spectrally unmixed pseudo-colored images were exported as .tif files using Phenochart. 

### 2.3. Quantitative Real-Time RT-PCR (qRT-PCR) Analysis

The cells were seeded into a 6-well plate (1 × 10^5^/well). After 48 h, the cells were washed three times with Dulbecco’s phosphate-buffered saline (DPBS; Gibco, Carlsbad, CA, USA, Cat# 14190-144). Total RNA was isolated using the RNeasy mini kit (QIAGEN, Hilden, Germany, Cat# 74104) following the manufacturer’s instructions, and cDNA was prepared from 1 μg of RNA with the Applied Biosystems^TM^ High-Capacity cDNA Reverse Transcription Kit (Cat# 4368813, Thermo Fisher Scientific, Waltham, MA, USA,) using random hexamers to primer the reaction. The cDNA was diluted 1:10 in RNase-free water. TaqMan™ Gene Expression Assays for *GAPDH* (Primer Assay ID Hs02786624_g1, Cat# 4331182), *RAMP1* (Hs00195288_m1, Cat# 4331182) and *CALCRL* (Assay ID Hs00907738_m1, Cat# 4331182) were purchased from Thermo Fisher Scientific (Waltham, MA, USA). Triplicate reactions for three biological replicates were carried out for each cell line using the TaqMan^TM^ Gene Expression Master Mix (Cat# 4369016, Thermo Fisher Scientific, Waltham, MA, USA,) and run on an AriaMx Real-Time PCR System (Agilent Technologies, Santa Clara, CA, USA). The cycling conditions included an initial step at 50 °C for 2 min, pre-denaturation at 95 °C for 10 min, and 40 cycles at 95 °C for 15 s and at 60 °C for 1 min. The relative quantification of target genes was calculated using the comparative cycle threshold (CT) method (2^−ΔCT^) with genes normalized to *GAPDH*.

### 2.4. Sciatic Nerve Ligation and Behavior Assays in Cancer Models

#### 2.4.1. Sciatic Nerve Ligation Model

To test whether cancer induces anterograde transportation of CGRP from DRG neuronal cell bodies toward the cancer paw, 2 × 10^5^ HSC-3 cells were inoculated into the left hind paw of nude mice. Three weeks after HSC-3 inoculation, the left sciatic nerve was ligated using three ligature knots (8-0, S&T^®^ CH-8212 Neuhausen/Switzerland) [16]. In control mice, three weeks after inoculation of HSC-3 cells, a sham operation was carried out to expose the sciatic nerve, but the nerve was not ligated. Twenty-four hours after sciatic nerve ligation, the sciatic nerve, 2 mm proximal to the ligation, was collected as described [17]. The sciatic nerves were fixed in 10% neutral buffered formalin, and embedded in an optimal cutting temperature compound (OCT). A 10 µm section of the sciatic nerve was transversely cut and processed for immunohistochemistry staining using mouse anti-CGRP antibody (ab81887, Abcam, Cambridge, UK, RRID:AB_1658411) and rabbit anti-PGP9.5 antibody (NBP229420, Novus Biological, Littleton, CO, USA). Images were captured with a Zeiss Plan-Apochromat LSM700, 63×/1.40 Oil DIC M27 objective. NIH ImageJ software was used to measure the CGRP signal intensity in each axon. The control group consisted of 3 mice; sciatic nerve ligation group consisted of 4 mice. All mice survived across the experimental timeline.

#### 2.4.2. Facial Mechanical Nociception Assay in the Tongue Orthotopic Xenograft Cancer Model

The facial mechanical nociception assay was used to measure facial mechanical allodynia as previously described [18]. Two weeks prior to the assay, the mice were acclimated for 1 h in the testing room 2 times per week. Mouse oral squamous cell carcinoma (MOC2, 5 × 10^3^ cells) were inoculated into the tongue of 8-week-old C57BL/6J (Jackson Laboratory, strain # 000664, RRID:IMSR_JAX:000664). In ascending order, von Frey filaments ranging from 0.008 to 4 g force (11 filaments in total) were used to measure withdrawal responses to mechanical stimulation of the cheek. Each fiber was applied once to the cheek, defined by the area between the nose and the ear, below the eye. If the mouse was moving or the response was unclear to the researcher, the same von Frey filament was reapplied to the same area of the cheek 10 s after the first stimulus or until the mouse stopped moving. A 5 min interval was set between the applications of von Frey filaments of different intensities. The facial nociception score was reported as a numerical average of the 11 responses in the response categories: 0: no response; 1: detection, the mouse is aware of the filament that stimulates the face; the mouse turns its head slightly to the object; 2: reaction, the mouse turns its head away quickly, pulls it backward or reacts with a single face wipe; 3: escape/attack, the mouse quickly escapes from the object, attacks the object with its paw or mouth, or reacts with two facial swipes; 4: multiple facial grooming, the mouse responds to the filament simulation with more than three facial wipes continuously. Both the control and treatment groups consisted of 10 mice. All mice survived across the experimental timeline.

#### 2.4.3. Mechanical Nociception Assay in the Hind Paw Xenograft Cancer Model

The paw xenograft cancer model permits the measurement of mechanical nociception in the paw. Baseline mechanical withdrawal thresholds were measured prior to tumor cell inoculation. The left hind paws of NU/J mice, 4- to 6-week-old, were injected with 1 × 10^5^ HSC-3 in 20 μL of DMEM and matrigel (1:1) [19,20]. To assess mechanical nociception, the mice were placed on a platform with a metal mesh floor and acclimated for 1 h. The paw withdrawal threshold was measured with von Frey filaments (Stoelting, Wood Dale, IL, USA) according to the up–down method for rats published by Chaplan et al. in 1994 with modifications for mice [21,22]. The withdrawal threshold was defined as the gram-force sufficient to elicit left hind paw withdrawal. A positive response was recorded if the mice showed one of the following reactions: 1—quick paw withdrawal; 2—immediate flinch when the tip of the von Frey filament is removed; 3—digit extension; 4—paw lift and licking of the paw; 5—repeated flapping of the paw to the mesh; or 6—attempt to run to escape from the stimulation. The interval between two trials was 10 s. The cut-off value was 4 g to prevent mechanical injury to the paw. Both the control and the treatment groups consisted of 5 mice. All mice survived across the experimental timeline.

### 2.5. Single-Cell RNA-Sequencing (scRNA-Seq) of Oral Cancer Cells and Oral Cancer Xenografts

#### 2.5.1. Preparation of Single-Cell Samples from Cultured Cells

Single-cell suspensions from HSC-3 and OSC-20 cell cultures were prepared according to the 10× Genomics^®^ Single Cell Protocols Cell Preparation Guide. Briefly, the cells were cultured in 10 cm plates to 70–80% confluence for 48 h. The cell culture medium was removed and replaced with 1.5 mL of 0.25% trypsin-EDTA solution. The trypsin-EDTA solution was immediately discarded and replaced with fresh 0.25% trypsin-EDTA (3 mL). After incubation for 3 min at 37 °C, digestion was stopped by adding 10 mL DMEM. Cells were centrifuged at 250 rcf for 5 min. The cell pellets were resuspended in 1 mL DMEM, filtered through 30 µm cell strainers (Cat# 130-098-458, Miltenyi, Gaithersburg, MD, USA) into 2 mL Eppendorf tubes and centrifuged at 150 rcf for 3 min to pellet the cells. The cells were washed twice in 1 mL PBS with 0.04% BSA. The cell pellets were resuspended in 250–500 µL volumes at a concentration > 7 × 10^5^ cells/mL and filtered through 40 µm Flowmi™ Cell Strainers (Cat# H13680-0040, SP Bel-Art, Wayne, NJ, USA). The samples with cell viability of 90–95% were processed with the 10× Genomics Single Cell protocol.

#### 2.5.2. Preparation of Single-Cell Samples from Orthotopic Xenografts

The tumors were excised from the tongues of mice with orthotopic xenografts of HSC-3 and OSC-20 (Section 2.2.1). The tissue was cut into small pieces with a scalpel and the tissues were digested in 3 mL of Hanks balanced salt solution buffer (HBSS, free from Ca^2+^ and Mg^2+^) containing 4 mg/mL collagenase II (Cat# 17101015, Gibco, Carlsbad, CA, USA) and 4.6 mg/mL dispase II (Cat# 17105041, Gibco, Carlsbad, CA, USA) for 45 min at 37 °C with gentle agitation every 2–3 min. The digested tissue pieces were separated by pipetting up and down 10 times before filtration through a 70 µm strainer followed via centrifugation at 250 rcf for 5 min. The pellet was washed with HBSS containing Ca^2+^ and Mg^2+^, resuspended in red blood cell lysis buffer for 1–2 min and then centrifuged. Pelleted cells were washed twice with PBS with 0.04% BSA, filtered through a 70 µm strainer and centrifuged at 150 rcf for 3 min to pellet the cells. The cell pellets were resuspended in 150–200 µL volumes at a concentration > 7 × 10^5^ cells/mL and filtered through 40 µm Flowmi™ Cell Strainers (Cat# H13680-0040, SP Bel-Art, Wayne, NJ, USA). The cell samples with >80% viability were processed with the 10× Genomics Single Cell protocol.

#### 2.5.3. 10× Genomics Library Preparation and Sequencing

Single cells were encapsulated into emulsion droplets using a Chromium Controller (10× Genomics). Single-cell RNA-sequencing libraries were constructed using the 10× Genomics Chromium Single Cell 3’ v3.1 Reagent Kit (PN-1000268) according to the manufacturer’s protocol. Amplified cDNA was evaluated on an Agilent BioAnalyzer 2100 using a High Sensitivity DNA Kit (Agilent Technologies) and final libraries on an Agilent TapeStation 4200 using High Sensitivity D1000 ScreenTape (Agilent Technologies). Individual libraries were diluted to 2 nM and pooled for sequencing. Pools were sequenced with 100 cycle run kits (28 bp Read1, 10 bp Index1 10 bp Index2 and 91 bp Read2) on the NovaSeq 6000 Sequencing System (Illumina, San Diego, CA, USA). The data were demultiplexed and quality checked using CellRanger v. 7, and aligned to the GRCh38-2020-A human and mm10-2020-A mouse reference genomes.

### 2.6. Statistical Analysis

Statistical analysis was performed with GraphPad Prism v9.5.1 (GraphPad Software, LLC., Boston, MA, USA). 

## 3. Results

### 3.1. Oral Cancer Xenografts Are Innervated by CGRP-Expressing Neurons

The presence of CGRP-ir neurons in orthotopic oral cancer HSC-3 and OSC-20 xenografts was confirmed using multiplex immunofluorescence staining for CGRP, GAP43 (marker of sprouting neurons), SOX10 (expressed in Schwann cell nuclei), PECAM (blood vessel marker), LYVE1 (lymphatic vessel marker) and TRP63 isoforms (epithelial cells) (Figure 1). Low-power views demonstrate dense growth of the cancer within and displacement of the normal tongue musculature, consistent with tongue squamous cell carcinoma in patients. Higher power views show GAP43-ir, SOX10-ir and CGRP-ir nerves and fine neurites. Blood vessels (PECAM-ir) and lymphatic channels (LYVE1-ir) course throughout. In OSC-20 xenografts, the CGRP-ir neurons, neurites, blood and lymphatic vessels course together through the stroma, a spatial relationship that recapitulates histologic findings in human oral squamous cell carcinoma. Little stroma is evident in HSC-3 xenografts and CGRP-ir nerves and neurites appear in close proximity to the HSC-3 cancer cells. The different growth phenotypes can also be appreciated in hematoxylin and eosin stained sections in Appendix A.

### 3.2. CGRP Is Transported Anterograde along the Nerve Innervating the Cancer

The retrograde labeling of neurons innervating xenograft and allograft tongue cancer models have demonstrated that cancer induced increased expression of CGRP in trigeminal ganglion neurons [9]. We used a sciatic nerve ligation model to determine whether HSC-3 cancer in the paw was associated with increased transport of CGRP from the dorsal root ganglion to the peripheral site of cancer in the paw (Figure 2a). Twenty-four hours post ligation, the section of nerve proximal to the site of ligation was collected and immunofluorescence imaging showed co-localization of CGRP-ir and neuronal marker PGP9.5-ir in neurons within the axon (Figure 2b). Greater CGRP-ir signal intensity was measured in the ligated nerve from the cancer paw compared to the sham operated paw (Figure 2c), consistent with cancer induced increased neuronal secretion of CGRP as a source of CGRP in the cancer microenvironment.

### 3.3. The CGRP Receptor Antagonist, Olcegepant, Reduces Nociception in Mice with Human or Mouse Oral Cancer Xenografts or Orthotopic Allografts

The question of whether the human oral cancer cell line HSC-3 produced nociception through CGRP was tested using two different models. Prior to paw inoculation, the mice were tested every week for 3 weeks with the paw von Frey nociception assay to monitor the development of mechanical allodynia (Figure 3a). At 3 weeks following cancer inoculation, when the mice had developed nociception, a single dose of olcegepant (1 mg/kg) [23] or vehicle (5% of DMSO in 250 µL normal saline) was administered intraperitoneally. The mice were then tested with the paw von Frey filament assay at 1, 3, 6, 12, and 24 h after olcegepant was administered (Figure 3b). Olcegepant reduced mechanical cancer allodynia for at least 6 h. 

The antinociceptive effect of olcegepant in the orthotopic tongue cancer model was evaluated with the anatomically relevant facial von Frey nociception assay. At 2 weeks following inoculation of MOC2 cells into the tongue, olcegepant (1 mg/kg) or vehicle was administered intraperitoneally and facial nociception was measured at 1, 3, 6, 12, and 24 h after administration. A single dose of olcegepant reversed the facial nociception score compared to the vehicle for at least 12 h (Figure 3c). 

### 3.4. Expression of CGRP Receptor Components in Oral Cancer Cells and Orthotopic Xenografts

#### 3.4.1. Oral Cancer Cells in Culture Express RAMP1 and CALCRL

Expression of *RAMP1* and *CALCRL* in oral precancer and cancer cell lines was confirmed with qRT-PCR. Expression of *RAMP1* in oral cancer cell lines was higher in cancer cell lines than in the dysplastic cell line, DOK and expression was higher in the cancer cell lines and DOK than in HaCaT, a spontaneously immortalized non-tumorigenic skin cell line. (Figure 4a). In contrast, the expression of *CALCRL* was higher in HaCaT than in DOK and OSC-20 (Figure 4b). The expression of *RAMP1* was 100× greater than that of *CALCRL*.

#### 3.4.2. *Ramp1*- and *Calcrl*-Expressing Cells Are Enriched in Fibroblasts and Immune Cells in the Xenograft Microenvironment

To investigate the expression of the human and mouse CGRP receptor components, the 10× Genomics Loupe Browser was used to visualize and explore the expression of human genes in the cultured cell samples and human and mouse genes in the tumor masses dissected from the xenografts. 

The CGRP receptor components encoded by *RAMP1* and *CALCRL* are expressed in HSC-3 and OSC-20 cells grown on tissue culture plastic (Figure 5 and Appendix A). The cell lines differed in the percentage of cells that expressed the receptor components. Whereas the majority of HSC-3 cells expressed *RAMP1* (76.8%), *RAMP1* was expressed in less than half of the OSC-20 Cells (39.8%). Fewer cells expressed *CALCRL* (0.50% and 1.05% in HSC-3 and OSC-20, respectively). Co-expression of the two CGRP receptor components was rare. Less than 1% of the cells co-expressed *RAMP1* and *CALCRL* (0.43% and 0.48% in HSC-3 and OSC-20, respectively).

The unbiased clustering of the xenograft samples identified 16 clusters in each xenograft (Appendix A). To evaluate the expression of human (HSC-3 and OSC-20) genes in xenografts, human cancer cell line-derived cells were defined as those expressing the human epithelial marker gene, *COL17A1*. Most HSC-3 (95.24%) and OSC-20 (80.83%) cell barcodes from cultures grown on plastic expressed *COL17A1* (Figure 5a and Appendix A). In the xenograft samples, *COL17A1* was expressed in 51.5% (5028/9763) of cell barcodes in HSC-3 and 4.03% (407/10097) of cell barcodes in OSC-20 (Figure 5b and Appendix A). The difference in percent of cancer cell line-derived cells reflects the difference in histology of the xenografts. OSC-20 elaborates an abundant stromal response (Appendix A). *RAMP1* was expressed in ~50% of COL17A1-expressing cells in both HSC-3 and OSC-20 xenograft samples (Figure 5b). Less than 1% of *COL17A1*-expressing cells also expressed *CALCRL* (0.26% and 0.74% in HSC-3 and OSC-20, respectively) or *RAMP1* and *CALCRL* (0.22% and 0.49% in HSC-3 and OSC-20, respectively). A greater percentage of mouse epithelial cells, i.e., those expressing the mouse *Col17a1* genes, also expressed *Calcrl* in the HSC-3 and OSC-20 xenograft samples (29.5% (13/44 cells) and 29% (29/100 cells), respectively).

To investigate the cell types in the xenograft microenvironment that co-express *Ramp1* and *Calcrl*, the data were filtered for cells co-expressing the receptor components with marker genes for fibroblasts (*Pdgfrb*), lymphatic endothelial cells (*Prox1*, *Lyve1*), blood endothelial cells (*Vwf*), Schwann cells (*Plp1*) and immune cells (*Ptprc*, encoding CD45). Fibroblasts and immune cells made up a larger proportion of the cells compared to other cell types in the xenograft tumor microenvironment in both HSC-3 and OSC-20 (Figure 6a, Appendix A). Lymphatic endothelial cells (*Lyve1-*, *Prox1*-expressing cells) were also frequent in the HSC-3 microenvironment. In both HSC-3 and OSC-20 xenografts, *Ramp1*-expressing cells were frequent among fibroblasts and immune cells, while *Calcrl*-expressing cells were frequent among fibroblasts, immune cells and endothelial cells (Figure 6b,c). To investigate the co-expression of *Ramp1* and *Calcrl*, the data were filtered for cell barcodes that expressed *Ramp1* and *Calcrl*, which were defined as “*Ramp1* + *Calcrl*” cells (Figure 6d). In both HSC-3 and OSC-20 xenografts, fibroblasts and immune cells were enriched for the *Ramp1* + *Calcrl* cells (Figure 6e), suggesting that CGRP in the oral cancer microenvironment is likely to impact the functions of these cell types.

## 4. Discussion

The current findings demonstrated that the CGRP antagonist olcegepant was effective for reducing nociception associated with oral cancer. Two different oral cancer pain models and associated nociceptive assays were used to demonstrate the antinociceptive effect. This study focused on the oral cancer microenvironment since that it is the site of generation of oral cancer pain. The source of CGRP in the oral cancer microenvironment was likely to be from the extensive innervation by CGRP-expressing neurons and the cancer-stimulated anterograde axonal transport of CGRP. CGRP is known as a neuropeptide; however, other non-neuronal cells have been reported to express CGRP [24]. Nevertheless, the neuronal source of CGRP in the cancer microenvironment is further supported by observations in other cancer models that reported cancer-induced increased expression of CGRP in nociceptors innervating the cancer [9,25], as well as studies manipulating CGRP expression via genetic and pharmacologic ablation of CGRP-expressing sensory neurons [10,26]. McIlvried et al., produced two syngeneic mouse oral cancer models by injecting two mouse oral cancer cell lines (MOC-1 and MOC-2) into C57BL/6 wild-type mice [9]. The investigators used immunohistochemistry to measure CGRP expression in the anatomically relevant afferent neurons (i.e., DiI labeled afferents in the mandibular branches of the trigeminal ganglia) [9]. They showed an increase in CGRP expression in the trigeminal ganglia. Nagamine et al. produced a rat oral cancer model by injecting rat squamous cell carcinoma cells (SCC-158) into the mandibular gingiva [25]. The investigators then measured CGRP with immunohistochemistry and showed an increase in CGRP in the trigeminal ganglia [25]. In the current study, the question was whether CGRP is transported in an anterograde direction toward the cancer three weeks after cancer inoculation. Because of the technical challenges associated with surgical ligation of a branch of the trigeminal ganglion, namely accessing and ligating the mandibular branch which is in the bone or surrounded by muscle at the skull base, a paw cancer model was studied. We measured CGRP on the proximal side of the ligation, but not the distal side or in the ganglia.

Expression of *RAMP1* and *CALCRL* was confirmed in cultured oral cancer cells (https://www.proteinatlas.org/ENSG00000132329-RAMP1/cell+line#head_and_neck_cancer, accessed on 1 June 2023; https://www.proteinatlas.org/ENSG00000064989-CALCRL/cell+line#head_and_neck_cancer accessed on 1 June 2023 and [9]). By scRNA-seq, *RAMP1* is expressed in a greater number of HSC-3 or OSC-20 cells than *CALCRL*. Moreover, the relative abundance of human *RAMP1-* and *CALCRL*-expressing cells is maintained in the human cancer cells in the orthotopic xenografts. *RAMP1* functions in cancer cells may extend beyond signaling with *CALCRL*. There is a growing appreciation of the number of GPCR-RAMP1 interactions in addition to functioning with *CALCRL* [27,28]. Moreover, *RAMP1* in the absence of exogenous CGRP is an upstream activator of *YAP1,* leading to mouse fibroblast proliferation in a wound healing model [29]. *YAP1* is an overexpressed and amplified oral cancer oncogene [30,31], suggesting a role for *RAMP1* in oral cancer promotion.

Multiple cells within the oral cancer microenvironment expressed the CGRP receptor (CLR/RAMP1), with the percentages of cells being greater in fibroblasts and immune cells. Antagonizing CGRP signaling in vivo reduced the growth of lung, oral and melanoma allografts [9,10,26,32]. The growth suppression has been attributed to effects on cancer cells [10,11], reduced angiogenesis [32], immunomodulation, including increased numbers of tumor-infiltrating T cells [9], reversal of CD8^+^ T cell exhaustion [26] and lack of cytoprotective autophagy under nutrient starvation conditions [10]. These studies highlight the complex and myriad ways in which CGRP-signaling in different cell types could contribute to the individual heterogeneity of oral cancers.

In addition to growth promotion, all of the cell types expressing the CGRP receptor components within the oral cancer microenvironment could mediate the antinociceptive effect of CGRP antagonism. The acute reduction in nociception observed at 1 h following olcegepant administration would not be consistent with the effect of CGRP antagonism on cancer growth. Schwann cells might mediate the antinociceptive effect as CLR signals from Schwann cells produce CGRP-dependent nociception, including facial cutaneous allodynia, a component of migraine [33]. Fibroblasts in the cancer microenvironment express the CGRP receptor and could contribute to oral cancer pain. Genes differentially expressed in oral cancers from patients who reported high pain levels are enriched for functions in the extracellular matrix [12]. Proliferation of cancer-associated fibroblasts has been proposed to contribute to pancreatic cancer pain; duloxetine reduces the proliferation of cancer and the proliferation of the associated fibroblasts resulting in antinociception in a mouse model of pancreatic cancer [34]. 

Immune cells have both nociceptive and antinociceptive effects in the setting of oral cancer. We used athymic mice that lack T lymphocytes; macrophages and neutrophils in the mice function normally. Macrophages express CLR/RAMP1 that contributes to the neuroimmune response to bacterial activation of Na_v_1.8-expressing nociceptors activated by bacteria [35]. Activation of the CGRP receptor on macrophages reduces neutrophil recruitment. Neutrophils are a source of local opioids within the oral cancer microenvironment of female mice [36,37]. Macrophage-mediated reduction in neutrophil infiltration could cause nociception through disinhibition of an endogenous analgesic mechanism. The multiplex immunofluorescence images of the tongue xenograft show that CGRP-expressing neurons course along with blood and lymphatic vessels, supporting the role of peptidergic neurons in vasodilation and edema associated with neuroinflammation. Reversal of neuroinflammation might contribute to the antinociceptive effect. 

Peptidergic neurons that innervate the oral cavity are the most likely source of CGRP in the oral cancer microenvironment that contributes to oral cancer pain. CGRP released from peptidergic neurons has potential effects at three locations involved in the signaling and aversive sensation of oral cancer pain. (1) Noxious stimulation of peptidergic neurons innervating oral cancer leads to CGRP secretion in the periphery, which acts upon blood vessels, immune cells and Schwann cells to induce neuroinflammation. (2) At the site of the trigeminal ganglia, housed in Meckel’s cave in the middle cranial fossa, and centimeters away from the oral cavity, CGRP exerts effects on glial cells and other sensory cell bodies within the ganglia leading to further activation. (3) The primary-order neurons synapse with second-order neurons in the subnucleus pars oralis within the spinal trigeminal nucleus. Efferent activity of the primary-order neurons influences glutamatergic signaling within second-order neurons, which then progress via the ventral trigeminothalamic tract to the thalamus, producing the perception of pain at the site of the cancer. In addition, administration of olcegepant might impact expression of CGRP at one or more of these sites, which is consistent with a feedback mechanism. Receptor antagonism along with reduced expression of CGRP would likely enhance the antinociceptive effect of olcegepant, although this is not a question addressed in the current study.

This study demonstrates that CGRP contributes to oral cancer pain, suggesting the antagonism of CGRP signaling to treat oral cancer pain. This study also confirms that multiple cell types express the CGRP receptor, which is important for the potential use of small-molecule-CGRP antagonists for the treatment of oral cancer pain. Antagonists could influence CGRP-mediated effects at the aforementioned different sites—within and distant to the cancer. While trials of olcegepant to treat migraine pain were discontinued due to hepatic toxicity [38], CGRP receptor antagonists and function-blocking monoclonal antibodies with acceptable safety profiles have been approved for acute and preventive migraine treatment. They could be repurposed for the treatment of oral cancer pain [39,40]. The CGRP receptor functions from both the periphery and the endosome [41]. Thus, a delivery mechanism that both limits the activity of the drug beyond the peripheral site of cancer and targets specific cellular compartments is desirable. The superficial location and accessibility of oral cancer enables local drug delivery into the cancer microenvironment. 

## 5. Conclusions

Oral cancer pain, which is a major clinical challenge, is generated in the periphery. Antagonism of CGRP, a mediator of trigeminal pain and neurogenic inflammation, alleviates oral cancer pain in preclinical models. Moreover, the observed innervation of xenograft tumors by CGRP-ir nerves supports neurogenic secretion as the source of CGRP in the cancer microenvironment. Targeting CGRP expression or CGRP-expressing neurons is a promising approach for addressing oral cancer pain.

## Figures and Tables

**Figure 1 cells-12-01675-f001:**
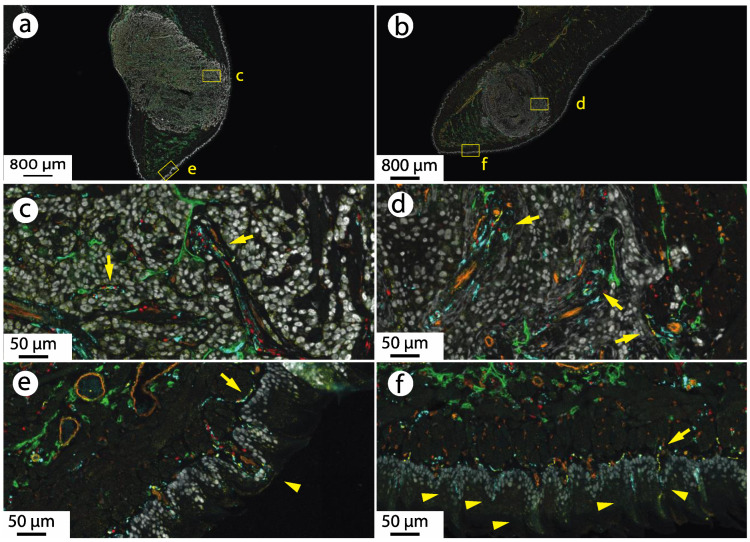
CGRP-expressing neurons innervating oral cancer xenograft tumors in the tongue. (**a**) HSC-3 tongue xenograft. The boxes indicate the area shown in (**c**,**e**). (**b**) OSC-20 tongue xenograft. The boxes indicate the area shown in (**d**,**f**). (**c**,**d**) Higher magnification images. Cancer cells (white, TRP63 isoforms, p63 + p40-ir) are innervated by bundles of neurons and neurites (yellow arrows). The neurons and neurites are identified by expression of GAP43-ir (teal), SOX10-ir (red, Schwann cells) and CGRP-ir (yellow). The nerve bundles run in close proximity to blood vessels (orange, PECAM-ir). Lymph vessels (green, LYVE1-ir) are also present. (**e**,**f**) Tongue dorsum. CGRP-ir and GAP43-ir nerves running with blood vessels below the epithelium innervate the fungiform (arrowhead in (**e**)) and filiform (arrowheads in (**f**)) papillae.

**Figure 2 cells-12-01675-f002:**
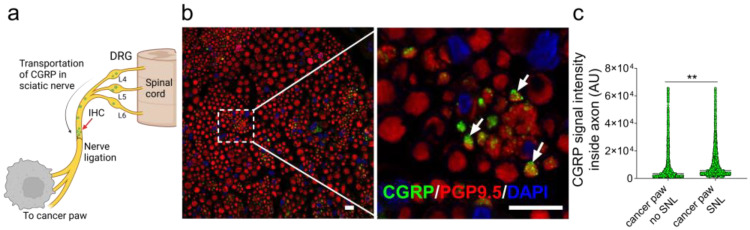
Accumulation of CGRP-ir in nerve fibers innervating oral cancer inoculated into the paw of mice. (**a**) The sciatic nerve was ligated distal to convergence of spinal nerves L4–L6 in mice 3 weeks after inoculation of HSC-3 cancer cells in the paw. The sciatic nerve sample was collected 24 h after ligation (red arrow). (**b**) Co-localization of CGRP-ir (green) inside the axons (PGP9.5-ir, red) of the sciatic nerve segment. Arrows indicate the colocalization of CGRP-ir (green) inside the axons (red). Scale bar in (**b**,**c**) = 10 µm. (**c**) CGRP-ir in the sciatic axons from sham operated mice (no SNL, *n* = 3 mice, *n* = 1829 axons) compared to sciatic nerve ligation mice (SNL, *n* = 4 mice, *n* = 2949 axons), t_(4776)_ = 4.88, ** *p* < 0.01, unpaired Student’s *t*-test.

**Figure 3 cells-12-01675-f003:**
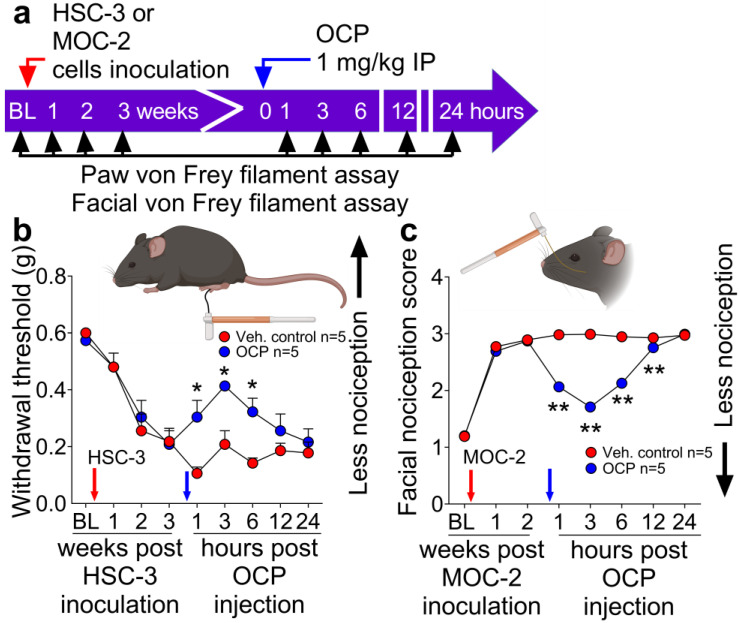
The CGRP receptor antagonist, olcegepant (OCP), reduces oral cancer nociception evoked by inoculation of oral cancer cells in the paw or tongue. (**a**) Experiment timeline indicating the baseline nociception measurement (BL) and the time at which HSC-3 or MOC2 cancer cells (red arrows) were inoculated into the left hind paw of NU/J mice or tongue of C57BL/6J mice, administration of olcegepant (OCP) or vehicle (blue arrows) and times of testing with the paw von Frey filament or facial nociception assay. (**b**) Paw cancer model. At 2 and 3 weeks after cancer cell inoculation (red arrow), the mice had developed cancer nociception. They were given OCP or vehicle (blue arrow) and tested for nociception with the paw von Frey filament assay. The direction indicating less nociception is indicated to the right of the plot (**c**) Orthotopic tongue cancer model. At 2 weeks after cancer cell inoculation (red arrow) the mice had developed nociception. They were given OCP or vehicle (blue arrow) and tested for nociception with the facial von Frey filament assay. The direction indicating less nociception is indicated to the right of the plot. In (**b**) F_(8,72)_ = 2.89, * *p* = 0.03, *n* = 5 mice in each group. In (**c**) F_(7,114)_ = 134.1, ** *p* < 0.0001, *n* = 10 mice in each group, OCP versus control at indicated time points by Two-way ANOVA.

**Figure 4 cells-12-01675-f004:**
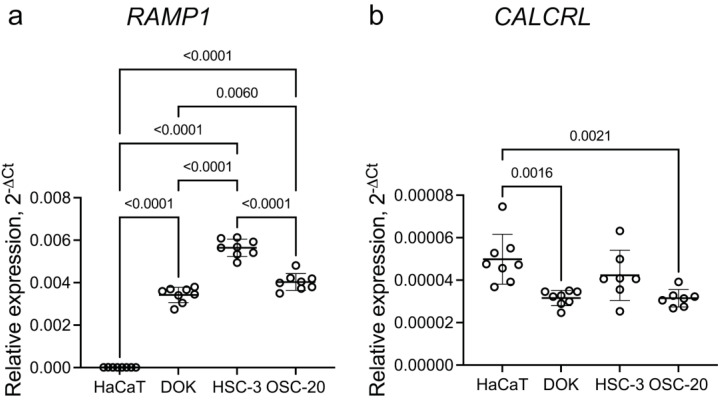
Expression of *RAMP1* and *CALCRL* in cell lines measured via qRT-PCR. (**a**) Expression of *RAMP1* relative to *GAPDH* was greater in dysplastic and cancer cell lines compared to non-tumorigenic skin keratinocyte line, HaCaT (**b**). The relative expression levels of *CALCRL* relative to *GAPDH* in HaCaT were greater than in DOK and OSC-20. Data are represented as mean ± SD. A one-way ANOVA, followed by Tukey’s multiple comparisons test, was carried out to evaluate statistical differences between groups.

**Figure 5 cells-12-01675-f005:**
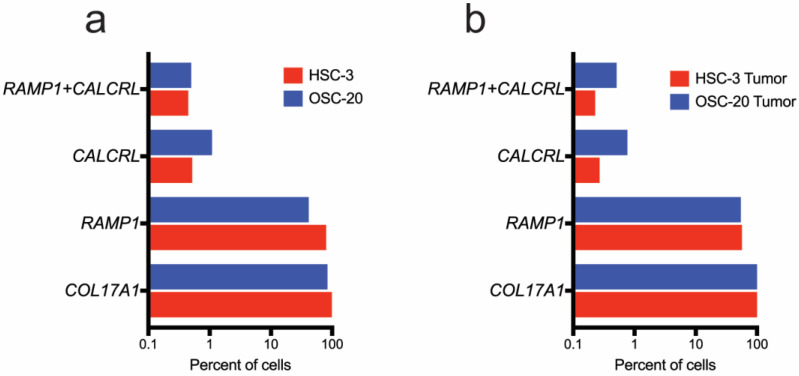
A greater number of HSC-3 and OSC-20 cells express *RAMP1* than *CALCRL*. Shown are the percent of cells expressing human CGRP receptor components and the epithelial marker gene, *COL17A1* in HSC-3 and OSC-20 cells grown (**a**) on tissue culture plastic and (**b**) as tongue xenografts. The relative frequencies of cells expressing human *RAMP1* and *CALCRL* were similar in tissue culture and in xenograft samples.

**Figure 6 cells-12-01675-f006:**
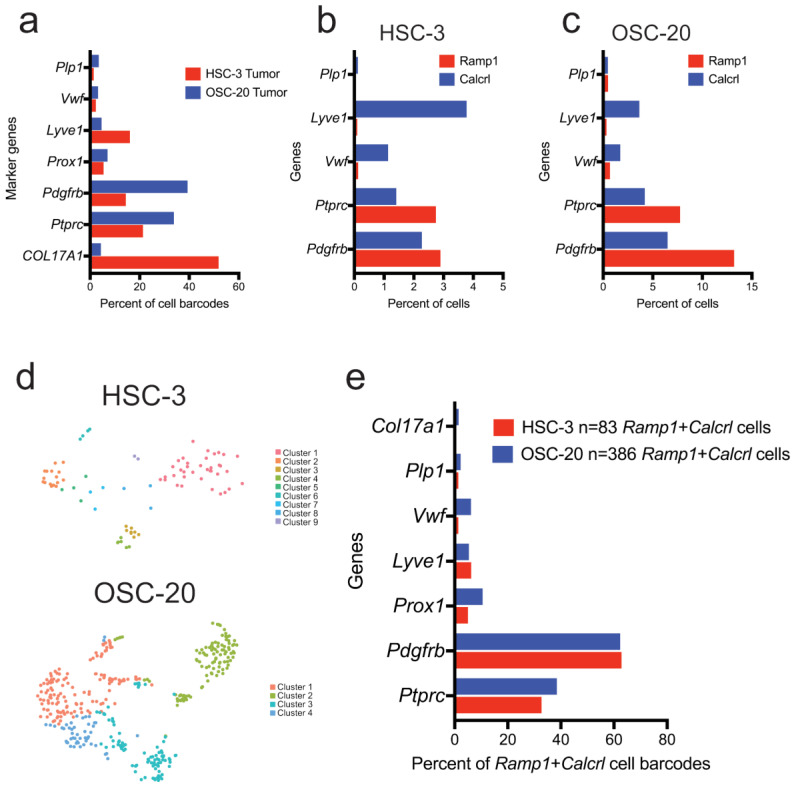
Cells expressing the CGRP receptor components were more frequent in fibroblast and immune cells in the HSC-3 and OSC-20 xenograft microenvironments. (**a**) Composition of the xenograft microenvironment for selected tissue types as defined by the expression of the marker genes for immune cells (*Ptprc*, encoding CD45), fibroblasts (*Pdgfrb*), lymphatic endothelial cells (*Prox1*, *Lyve1*), blood endothelial cells (*Vwf*) and Schwann cells (*Plp1*). (**b**,**c**) Expression of *Ramp1* and *Calcrl* in selected tissue types of the xenograft microenvironment as determined by co-expression with tissue marker genes. (**d**) t-SNE representation of cell barcodes co-expressing *Ramp1* and *Calcrl*—*Ramp1* + *Calcrl* cells. (**e**) Tissue type distribution of *Ramp1* + *Calcrl* cells in the xenograft microenvironment as defined by co-expression with tissue marker genes.

## Data Availability

The data presented in this study are available in the Appendix A.

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
