# Peer review of "Calcitonin Related Polypeptide Alpha Mediates Oral Cancer Pain"

_cells, 2023, doi:10.3390/cells12131675_

Round 1

Reviewer 1 Report

The authors present a nice study on the CGRP signaling mediating cancer pain in oral cancer. While this is an elaborated study with in vitro, in vivo and thorough analysis, this reviewer would like to stress out some points that can be improved.

1.     It is imperative to improve the statistical analysis description. The sample size, all the tests performed and refer if different tests were performed depending on the experimental dataset.

2.     It is described in the results that the size of the tumor developed by HSC differs from OSC. Is this due to the proliferative capacity of the cells? Or efficiency on other cells recruitment? Please include data to support your reply along with H&E stainings.

3.     How is the innervation profile in the healthy tissue? This information must be included for a better perspective on the innervation profile/modulation by the cancer.

4.     In fig.2 the authors indicate an accumulation of CGRP in nerves. Is there an increase of CGRP is the DRG for the HSC model? How was the expression on the distal side of the ligation?

5.     It was not clear why the authors evaluated the CGRP expression in DRG rather than TG. How is the CGRP expression in the TG from animals with and without oral cancer?

6.     The co-localization appears to be the nerve fibers itself in transversal cut, rather than CGPR inside neurons. The images presented do not have enough resolution to support the description of the authors. Better images are needed or the statement should be modified.

7.     Apart from the sciatic nerve ligation model, information about sample size, experimental groups, number of animals, survival across the experimental timeline, etc are not described (must be added to M&M and results sections). Accordingly, in fig.3 the number of animals tested and the respective SD of the data presented (3c) must be included.

8.     The cells were characterized to identify the main population expressing CGRP receptors, showing a higher enrichment on fibroblasts and immune cells. If the antagonist OCP is acting on the fibroblasts/immune cells. Did the authors investigate if there is a feedback loop on CGRP expression by the nerves? Meaning, if the administration of the OCP impacts the expression of CGRP at DRG or central levels?

9.     In line 382 it is not clear if the evaluation was made only in vitro as the authors refer to “genes in xenografts” and then “cultures grown on plastic expressed”.

10.  In line 386, the authors refer again to xenograft samples. In this case, the % of CGRP receptors by human cancer cells is very low. This reviewer would like to understand how are the levels in the remaining cells without the xenograft, meaning the control tissues are missing.

Minor:

Line103 – indicate here the description for the mouse MOC2 cell line (it appears afterwards);

Figure 1 – Following the description of the results in the text, the authors could consider presenting first on the left the HSC and on the right the OSC. The images presenting the same mag. should be of the same size. For all, the scale bar must be included. In the caption, please refer to the scale size instead of the magnification used.

Author Response

Reviewer #1

The authors present a nice study on the CGRP signaling mediating cancer pain in oral cancer. While this is an elaborated study with in vitro, in vivo and thorough analysis, this reviewer would like to stress out some points that can be improved.

  1. It is imperative to improve the statistical analysis description. The sample size, all the tests performed and refer if different tests were performed depending on the experimental dataset.

The requested information, including sample size and statistical test, has been added to the figure legends.

  1. It is described in the results that the size of the tumor developed by HSC differs from OSC. Is this due to the proliferative capacity of the cells? Or efficiency on other cells recruitment? Please include data to support your reply along with H&E stainings.

In response to comments #2 and #3, we prepared a new Figure 1.

In the paragraph beginning at line 382 of the original document, now line 391, we noted that the HSC-3 and OSC-20 tumor samples differed in the percentage of cells that expressed COL17A1. The tumor masses of both cell lines similarly fill the anterior tongue and displace the muscle tissue (new Figure 1a and b). The heterogeneous phenotype of the xenograft tumors from oral cancer cell lines has been reported (PMID 25765182). The OSC-20 tumors are characterized by a prominent stromal component. Therefore, the single cell preparation from OSC-20 tumors contains a higher proportion of stromal cells than HSC-3 preparations with the result that human epithelial cells make up a smaller proportion of the total number of cells.

We have included the requested hematoxylin and eosin stained images of the HSC-3 and OSC-20 xenografts, which show the different growth phenotypes, in Figure S1.

  1. How is the innervation profile in the healthy tissue? This information must be included for a better perspective on the innervation profile/modulation by the cancer.

We have included an image of the normal epithelium showing the blood, vessels and nerves that course below the epithelium and enter a fungiform papilla in new Figure 1e and filiform papillae in new Figure 1f.

  1. In fig.2 the authors indicate an accumulation of CGRP in nerves. Is there an increase of CGRP is the DRG for the HSC model? How was the expression on the distal side of the ligation?

The purpose of the sciatic nerve ligation model was to test whether oral cancer induces anterograde transportation of CGRP from DRG neuronal cell bodies toward the paw with cancer. We did not evaluate expression of CGRP in the DRG or distal to the site of nerve ligation. Measurement of CGRP expression in the DRG addresses an important question, which is whether cancer induces expression at the ganglia. See the response to question #5 below. To address the reviewer’s question #4, we added a statement in the Discussion.

  1. It was not clear why the authors evaluated the CGRP expression in DRG rather than TG. How is the CGRP expression in the TG from animals with and without oral cancer?

McIlvried et al., (PMID 35388989) produced two syngeneic mouse oral cancer models by injecting two mouse oral cancer cell lines (MOC-1 and MOC-2) into the tongue of C57BL/6 wild type mice. The investigators used immunohistochemistry to measure CGRP expression in the anatomically relevant afferent neurons (i.e., DiI labeled afferents in the mandibular branch of the TG). They showed an increase in CGRP expression in the trigeminal ganglia (Figures 2 D and E). Nagamine et al., (PMID 16942952) produced a rat oral cancer model by injecting rat squamous cell carcinoma cells (SCC-158) into the mandibular gingiva. The investigators then measured CGRP with immunohistochemistry and showed an increase in CGRP in the trigeminal ganglia (Figure 5A, Tables 2 and 3). In the current study, the question was whether CGRP is transported in an anterograde direction toward the cancer three weeks after cancer cell inoculation. The paw cancer model was studied, because there are technical challenges associated with surgical ligation of a branch of the trigeminal ganglion, namely, the mandibular branch is in bone or surrounded by muscle at the skull base.

To address the reviewer’s important point a reference to cancer-induced CGRP expression in the trigeminal ganglia, relative to our approach, has been added to the Discussion.

  1. The co-localization appears to be the nerve fibers itself in transversal cut, rather than CGPR inside neurons. The images presented do not have enough resolution to support the description of the authors. Better images are needed or the statement should be modified.

New images, which have higher resolution, have been generated to replace Figure 2 b and c. The images show CGRP inside the neurons.

  1. Apart from the sciatic nerve ligation model, information about sample size, experimental groups, number of animals, survival across the experimental timeline, etc are not described (must be added to M&M and results sections). Accordingly, in fig.3 the number of animals tested and the respective SD of the data presented (3c) must be included.

The information requested has been added to the Materials and Methods section and the figure legends for Figure 2 and 3.

  1. The cells were characterized to identify the main population expressing CGRP receptors, showing a higher enrichment on fibroblasts and immune cells. If the antagonist OCP is acting on the fibroblasts/immune cells. Did the authors investigate if there is a feedback loop on CGRP expression by the nerves? Meaning, if the administration of the OCP impacts the expression of CGRP at DRG or central levels?

The reviewer raises an interesting question whether there is a feedback loop. We are unaware of a study that addressed the question, and we did not investigate it. We added the following to the Discussion at line 523.

In addition, administration of olcegepant might impact expression of CGRP at one or more of these sites, consistent with a feedback mechanism. Receptor antagonism along with reduced expression of CGRP would likely enhance the antinociceptive effect of olcegepant, although this is not a question addressed in the current study.

  1. In line 382 it is not clear if the evaluation was made only in vitro as the authors refer to “genes in xenografts” and then “cultures grown on plastic expressed”.

We clarified the statement to indicate that both cells in vitro and tumors from the xenografts were studied.

“To investigate the expression of the human and mouse CGRP receptor components, we used the 10x Genomics Loupe Browser to visualize and explore expression of human genes in the cultured cell samples and human and mouse genes in the tumor masses dissected from the xenografts.”

  1. In line 386, the authors refer again to xenograft samples. In this case, the % of CGRP receptors by human cancer cells is very low. This reviewer would like to understand how are the levels in the remaining cells without the xenograft, meaning the control tissues are missing.

In response, we added the requested information on the number of mouse epithelial cells identified by expression of the mouse gene, Col17a1 in the xenograft samples that also express the mouse Calcrl gene.

“Less than 1% of COL17A1 expressing cells also expressed CALCRL (0.26% and 0.74% in HSC-3 and OSC-20, respectively) or RAMP1 and CALCRL (0.22% and 0.49% in HSC-3 and OSC-20, respectively). A greater percentage of mouse epithelial cells, i.e., those expressing the mouse Col17a1 genes, also expressed Calcrl in the HSC-3 and OSC-20 xenograft samples (29.5% (13/44 cells) and 29% (29/100 cells), respectively).”

Minor:

Line103 – indicate here the description for the mouse MOC2 cell line (it appears afterwards);

The following description of the MOC2 cell line has been added at line 105 in the revised version of the manuscript:

Mouse cell line MOC2 (RRID:CVCL_ZD33, passage 5), obtained from Dr. R. Uppaluri [15], was derived from a primary tumor induced by dimethylbenz(a)anthracene (DMBA) carcinogen treatment in a Cxcr3 deficient mouse on a C57BL/6 background. The MOC2 cells are aggressive and metastasize to regional lymph nodes. The MOC2 cells were cultured according to the recommended protocol provided by Kerafast, Inc.

Figure 1 – Following the description of the results in the text, the authors could consider presenting first on the left the HSC and on the right the OSC. The images presenting the same mag. should be of the same size. For all, the scale bar must be included. In the caption, please refer to the scale size instead of the magnification used.

The recommended changes have been made in the new Figure 1.

Thank you for considering our manuscript for publication in Cells.

Sincerely,

Brian L. Schmidt, DDS, MD, PhD

Reviewer 2 Report

I have nothing to add. accept as is.

Author Response

Thanks for your comments

Reviewer 3 Report

It is an interesting manuscript that shows evidence of the action of CGRP as a mediator of oral cancer pain and suggests that using a CGRP antagonist (olcegepant) might help alleviate oral cancer pain. Identifying peripheral CGRP-ir neurons innervating oral cancer cells and CGRP receptor expression in other cells within the oral cancer microenvironment (immune cells, fibroblasts, Schwann cells, and endothelial cells) is worth mentioning. These aspects coincide with the primary goals of this special issue. Additionally, no red flags were detected in terms of similarities and paraphrasing. However, some aspects need to be addressed in the manuscript to be publishable.

1. The authors should consider including other references at the end of the Introduction and not leave the reader with the idea that only one article published thus far correlates CRP with pain associated with oral cancer (see line 87). For example, the following references can be added at the end of the sentence.

Sensory Neurotransmitter Calcitonin Gene-Related Peptide Modulates Tumor Growth and Lymphocyte Infiltration in Oral Squamous Cell Carcinoma.

McIlvried LA, et al. Adv Biol (Weinh). 2022. PMID: 35388989

Calcitonin gene-related peptide: A promising bridge between cancer development and cancer-associated pain in oral squamous cell carcinoma.

Zhang Y, et al. Oncol Lett. 2020. PMID: 32994816.

The neuropeptide calcitonin gene-related peptide links perineural invasion with lymph node metastasis in oral squamous cell carcinoma.

Zhang Y, et al. BMC Cancer. 2021. PMID: 34800986.

In fact, the third reference is already part of the cited bibliography.

2. Regarding statistics, the Materials and Methods section only mentions using GraphPad Prism v9.5.1. None of the analyses are indicated.

3. Are the titles of Figures 2 and 3 appropriate? Figure 2. Accumulation of CGRP-ir in nerve fibers innervating oral cancer in mice. Figure 3. The CGRP receptor antagonist, olcegepant (OCP), reduces oral cancer nociception.

Although the findings reported in the paw cancer pain model are very interesting, their inclusion in this study should be better explained. In Figure 2, the legend and images show an interesting colocalization of CGRP-ir exclusively within sciatic nerve axons. Combining the results of the paw model with those of the oral model in the same figure, with a title related solely to oral cancer pain, might sound distracting.

Please consider that the rating scales of the observed behavioral changes in both anatomical locations are not similar, and their graphical representation shows results that point in different directions (although the effect of the antagonist is evident).

4. The authors used two strains of mice, nude mice (NU/J and C57BL/6). The reason for their differential use should appear in the Materials and Methods section.

5. The Animal Care and Use Committee certification appears at a different location from where it should be.

1. It seems that more than one author wrote the text. Standardizing writing styles may be convenient.

2. The authors should avoid repeated personalization in sections such as the Abstract and Discussion (i.e., we demonstrated, we used, we focused, we found, we confirmed, we showed, our study). In addition, connecting the sentences in the Conclusions section will make the paragraph more fluid.

Author Response

Reviewer #2

Comments and Suggestions for Authors

It is an interesting manuscript that shows evidence of the action of CGRP as a mediator of oral cancer pain and suggests that using a CGRP antagonist (olcegepant) might help alleviate oral cancer pain. Identifying peripheral CGRP-ir neurons innervating oral cancer cells and CGRP receptor expression in other cells within the oral cancer microenvironment (immune cells, fibroblasts, Schwann cells, and endothelial cells) is worth mentioning. These aspects coincide with the primary goals of this special issue. Additionally, no red flags were detected in terms of similarities and paraphrasing. However, some aspects need to be addressed in the manuscript to be publishable.

  1. The authors should consider including other references at the end of the Introduction and not leave the reader with the idea that only one article published thus far correlates CRP with pain associated with oral cancer (see line 87). For example, the following references can be added at the end of the sentence.

To our knowledge the McIlvried et al. paper (reference 9) is the only one that assessed CGRP expression and patient reported pain. By contrast, Zhang et al. (references 10, 11) studied innervation of oral cancers by nociceptors, defined by expression of CGRP or TRPV1. While they showed that oral cancers are innervated by CGRP+ and TRPV1+ nerves and that patients with a higher proportion of CGRP+ nerves had worse survival, they did not measure patient reported pain or nociceptive behavior in animal models.

  1. Regarding statistics, the Materials and Methods section only mentions using GraphPad Prism v9.5.1. None of the analyses are indicated.

The method of analyses is indicated in each of the figure legends.

  1. Are the titles of Figures 2 and 3 appropriate? Figure 2. Accumulation of CGRP-ir in nerve fibers innervating oral cancer in mice. Figure 3. The CGRP receptor antagonist, olcegepant (OCP), reduces oral cancer nociception.

Although the findings reported in the paw cancer pain model are very interesting, their inclusion in this study should be better explained. In Figure 2, the legend and images show an interesting colocalization of CGRP-ir exclusively within sciatic nerve axons. Combining the results of the paw model with those of the oral model in the same figure, with a title related solely to oral cancer pain, might sound distracting.

All of the findings presented in Figure 2 are from a paw cancer model that was inoculated with an oral cancer cell line. To clarify, the legend has been changed to, “Accumulation of CGRP-ir in nerve fibers innervating oral cancer inoculated into the paw of mice.”

The findings presented in Figure 3 are from two different models, injection of oral cancer cells into the paw or the tongue. To clarify, the legend has been changed to, “The CGRP receptor antagonist, olcegepant (OCP), reduces oral cancer nociception evoked by inoculation of oral cancer cells into the paw or tongue.”

Please consider that the rating scales of the observed behavioral changes in both anatomical locations are not similar, and their graphical representation shows results that point in different directions (although the effect of the antagonist is evident).

In Figure 3 the direction indicating less nociception is indicated to the right of each of the plots. For clarification and to address the reviewer’s concern the following statement has been added to the figure legend, “For (b) and (c) direction indicating less nociception is indicated to the right of each of the plots.”

  1. The authors used two strains of mice, nude mice (NU/J and C57BL/6). The reason for their differential use should appear in the Materials and Methods section.

The following statement that clarifies the use of nude mice and has been added to the Materials and Methods, “…which are required for generation of a xenograft.”

  1. The Animal Care and Use Committee certification appears at a different location from where it should be.

The statement has been moved to the end of section 2.2.1. Generation of mouse orthotopic xenografts, which is the first reference to mice.

Comments on the Quality of English Language

  1. It seems that more than one author wrote the text. Standardizing writing styles may be convenient.

The text in the Abstract, Discussion and Conclusion, along with one sentence in the Introduction, has been modified.

  1. The authors should avoid repeated personalization in sections such as the Abstract and Discussion (i.e., we demonstrated, we used, we focused, we found, we confirmed, we showed, our study). In addition, connecting the sentences in the Conclusions section will make the paragraph more fluid.

All statements throughout the manuscript written in the first person have been edited to a passive form.

The Conclusions section now reads as follows:

Oral cancer pain, which is a major clinical challenge, is generated in the periphery. Antagonism of CGRP, a mediator of trigeminal pain and neurogenic inflammation, alleviates oral cancer pain in preclinical models. Moreover, the observed innervation of xenograft tumors by CGRP-ir nerves supports neurogenic secretion as the source of CGRP in the cancer microenvironment. Targeting CGRP expression or CGRP expressing neurons is a promising approach for addressing oral cancer pain.

Thank you for considering our manuscript for publication in Cells.

Sincerely,

Brian L. Schmidt, DDS, MD, PhD

Round 2

Reviewer 1 Report

The authors have addressed the issues raised by this reviewer.

There are 2 refs (lines 477, 482) that are not numbered.

Thank you

Author Response

We appreciate that the reviewer states that we have addressed the issues raised. In response to the minor comments we have added the references to lines 477 and 482. Thank you, Brian Schmidt